# Lymphocyte Phenotypes and Protein-Bound Uremic Toxins as Determinants of Clinical Outcomes in Hemodialysis Patients

**DOI:** 10.3390/ijms262110376

**Published:** 2025-10-24

**Authors:** Theodoros Tourountzis, Georgios Lioulios, Stamatia Stai, Steven Van Laecke, Eleni Moysidou, Michalis Christodoulou, Ariadni Fouza, Asimina Fylaktou, Konstantia Kantartzi, Griet Glorieux, Maria Stangou

**Affiliations:** 1Protypo Dialysis Center of Thessaloniki, 55535 Thessaloniki, Greece; ttourou@gmail.com; 2Department of Nephrology, School of Medicine, Aristotle University of Thessaloniki, General Hospital “Hippokratio”, 54642 Thessaloniki, Greece; pter43@yahoo.gr (G.L.); staimatina@yahoo.gr (S.S.); moysidoueleni@yahoo.com (E.M.); michalischristodoulou22@gmail.com (M.C.); ariadnefou@gmail.com (A.F.); 3Department of Internal Medicine and Pediatrics, Nephrology Unit, Ghent University Hospital, 9000 Gent, Belgium; steven.vanlaecke@ugent.be (S.V.L.); griet.glorieux@ugent.be (G.G.); 4Department of Immunology, National Peripheral Histocompatibility Center, General Hospital “Hippokratio”, 54642 Thessaloniki, Greece; fylaktoumina@gmail.com; 5Department of Nephrology, University Hospital of Alexandroupolis, Democritus University of Thrace, 68100 Alexandroupolis, Greece; kkantart@med.duth.gr

**Keywords:** protein bound uremic toxins, immunosenescence, morbidity, mortality, hemodialysis

## Abstract

The impact of protein bound uremic toxins (PBUTs) and lymphocyte alterations in morbidity and mortality in patients on hemodialysis (HD) is of great concern. The aim of this study was the assessment of association between PBUTs, immunosenescent lymphocytes’ phenotype and clinical events [cardiovascular, severe infections (hospitalization due to infection, respiratory infection), all-cause mortality] during 2-year follow-up. In this prospective observational study, lymphocytes’ phenotype of 54 patients on HD and 31 age-matched controls was analyzed by flow cytometry, and simultaneously, PBUT serum levels [hippuric acid (HA), indoxyl sulfate (IxS), p-cresyl sulfate (pCS), p-cresyl glycuronide (pCG), in-dole-3-acetic acid (IAA), and 3-carboxy-4-methyl-5-propyl-2-furanpropionate (CMPF)] were quantified by ultra-performance liquid chromatography. Patients with increased levels of free IxS and total and free HA had higher mortality within a 2-year follow-up period (*p* = 0.049, *p* = 0.01, *p* = 0.01, respectively). In patients who experienced cardiovascular events, higher concentrations of CMPF (*p* = 0.015) were observed. Higher total and free HA levels associate with increased all-cause mortality in patients on HD, independently of age, dialysis vintage, and decreased count of CD4+CD45RA+CD31+ and naïve B cells (CD19+IgD+CD27−). In patients on HD, increased levels of total and free HA associate with an increased risk of death.

## 1. Introduction

Patients undergoing hemodialysis (HD) experience increased morbidity and mortality. Several factors correlated with elevated all-cause mortality, including advanced age, diabetes mellitus, cardiovascular disease, and increased C-reactive protein levels [1]. The elevated risk of cardiovascular disease in patients undergoing HD are due to hypervolemia, uremic cardiomyopathy, secondary hyperparathyroidism, lipid disorders, anemia, and the accumulation of protein bound uremic toxins (PBUTs) [2]. Moreover, the utilization of central venous catheter at the commencement of HD is a significant factor associated with hospitalizations resulting from infections of any origin, bacteremia, and infections related to HD access [3].

The European Uremic Toxin Work Group categorized uremic toxins into three groups, based on their physicochemical properties and their protein binding: (1) water-soluble low molecular weight molecules (under 500 Daltons), (2) middle molecules (500 Daltons or higher), and (3) protein bound solutes. The latter, also known as PBUTs, include substance among others, like indoxyl sulfate (IxS), p-cresyl sulfate (pCS), p-cresyl glycuronide (pCG), hippuric acid (HA), indole-3-acetic acid (IAA), and 3-carboxy-4-methyl-5-propyl-2-furanpropionate (CMPF), all of which have low molecular weight [4,5]. Recent advancements in HD membranes and other removal techniques, have refined this classification [6]. Due to their strong protein binding, current methods of HD are not sufficiently effective in removing PBUTs. Consequently, PBUTs accumulate in patients on HD, correlating with high comorbidity and mortality [7]. Uremic toxins are linked to kidney and cardiovascular damage, endothelial dysfunction, elevated infection susceptibility, intestinal dysbiosis, and cognitive impairment, potentially resulting in higher mortality and diminished quality of life for individuals with chronic kidney disease (CKD) [6,8,9,10].

Immunosenescence, characterized by dysregulations within both the innate and adaptive immune systems, is linked to the aging process. This contributes to increased susceptibility to infections, an elevated risk of malignancy and a poor response following vaccination [11]. Moreover, the senescence of T cells is related with low-grade inflammation, referred to as inflammaging, and both factors are implicated in the pathogenesis of age-related conditions, including cardiovascular disease, cancer, and neurodegenerative disorders [12]. Patients with end-stage kidney disease (ESKD) exhibit a premature senescence of the immune system which is associated with other comorbidities. In patients undergoing HD, potential immunosenescent alterations, include reduced number of T cells, naïve T cells, and CD4^+^ naïve T cells, along with a decreased percentage of T cells and an increased percentage of CD8^+^ central-memory T cells. The decreased count of naïve T cells, may significantly have an impact on all-cause mortality in patients undergoing HD [13].

The main aim of this study was the evaluation of association between PBUT serum levels, lymphocytes’ phenotype, and clinical outcome including cardiovascular events, severe infections (hospitalization due to infection, respiratory infection) and all-cause mortality, in patients on HD, during a 2-year follow-up period.

## 2. Results

### 2.1. Patients’ Characteristics

Fifty-four patients on HD and thirty-one healthy age-matched controls were included in the study. The mean age of the patient group (23 females, 31 males) was 51.3 ± 16.9 years and of the control group (15 females, 16 males) it was 51.3 ± 17.2 years (*p* = 0.993). The median HD vintage was 67 (20.7–95.2) months, 24/54 (44.4%) patients were on online hemodiafiltration, and 30/54 (55.6%) on HD. Mean body mass index was at upper normal limits, 24.5 ± 3.6 kg/m^2^. Dialysis-related details, comorbidities, and causes of ESKD are depicted in Table 1. The low percentage of cardiovascular comorbidity may be explained due to the fact that patients with high comorbidity were excluded, such as those with diabetes mellitus, and the patients were relatively young.

Laboratory parameters of the patients and control group are shown on Table 2.

### 2.2. PBUT and Peripheral Lymphocytic Phenotype of Patients on HD

Increased levels of serum PBUTs were observed in patients compared with control group (Table 3).

Patients on HD had increased white blood cells compared to control group; 7100 (5500–8325) vs. 6200 (5300–7100) cells/μL (*p* = 0.046). Moreover, reduction in CD4+ [679.5 (483–862.2) vs. 999 (786–1237) cells/μL (*p* < 0.001)], and CD8+ cells [377 (261.7–531.5) vs. 451 (296–746) cells/μL (*p* = 0.13)] was found in patients on HD. Furthermore, reduced numbers of B cells and their subpopulations were found in patients on HD in comparison to healthy individuals, namely: CD19+ 91 (52.2–131.2) vs. 248 (163–388) cells/μL (*p* < 0.001); naïve (IgD+CD27-) 56.5 (27.7–96.2) vs. 144 (91–258) cells/μL (*p* < 0.001); IgM memory (IgD+CD27+), 5 (3–10.2) vs. 23 (11–32) cells/μL (*p* < 0.001); switched memory (IgD-CD27+) 13 (7.7–18.5) vs. 35 (22–61) cells/μL (*p* < 0.001); double negative (IgD-CD27-) 7 (4.7–12) vs. 26 (14–43) cells/μL (*p* < 0.001), (Table 4).

Patients treated with online hemodiafiltration had slightly higher total white blood cells, neutrophils, and lymphocytes compared to those on conventional HD [7150 (5775–8550) vs. 7000 (5275–7600) cells/μL (*p* = 0.3), 4650 (3525–5875) vs. 4400 (3475–5400) cells/μL (*p* = 0.47), 1500 (1325–1800) vs. 1400 (1050–1825) cells/μL (*p* = 0.346), respectively], but these results were not statistically significant. Patients with and without residual renal function did not have statistically significant changes in the total numbers of white blood cells, neutrophils, and lymphocytes [6900 (5450–7800) vs. 7100 (5600–8500) cells/μL (*p* = 0.619), 4500 (3400–5200) vs. 4600 (3750–5850) cells/μL (*p* = 0.267), 1600 (1300–1900) vs. 1400 (1000–1750) cells/μL (*p* = 0.105), respectively], or in the lymphocyte phenotype. Additionally, lymphocytes did not correlated significantly with other parameters of dialyzability, such as Kt/V or dialysis membrane surface area.

### 2.3. Association with Clinical Events

#### 2.3.1. Association of Immune Phenotype with Mortality and Morbidity

Patients on HD who died in the 2-year follow-up (n = 6), compared to those remained alive (n = 48), had significantly reduced count of naïve B cells (CD19+IgD+CD27-) [23.5 (11.25–54.25) vs. 58.5 (30–98.75), *p* = 0.032, respectively], but no significant changes in the rest subpopulations of T lymphocytes, and also no differences in B lymphocytes. In addition, patients’ morbidity, including cardiovascular events or severe infections, were not associated with differences in baseline lymphocyte subtype.

#### 2.3.2. Multiple Regression Analysis

Dialysis vintage was positively correlated with increased levels of total (r = 0.379, *p* = 0.005) and free HA (r = 0.392, *p* = 0.003), but not with IxS (r = 0.252, *p* = 0.066). Additionally, significant effect of increased levels of total and free HA in mortality remained in Cox regression analysis, either examined alone as continuous variables [*p* = 0.023, 95% confidence interval (CI) 1.024–1.366, *p* = 0.008, 95% CI 1.077–1.628, respectively] or in combination with age, dialysis vintage, and total number of CD4+CD45RA+CD31+ and CD19+IgD+CD27- lymphocytes (*p* = 0.041, 95% CI 1.007–1.444, *p* = 0.01, 95% CI 1.093–1.948, respectively). In contrary, the significance of the association between increased levels of free IxS and mortality disappeared after being analyzed with Cox regression with the above-mentioned parameters.

#### 2.3.3. Association of PBUTs with Mortality and Morbidity

Log-rank tests were run to determine if there were differences in the clinical events distribution (cardiovascular events, hospitalization due to infection, and all-cause mortality) for the PBUTs that were examined. Kaplan–Meier survival analysis tests found that patients’ survival was significantly different when they were clustered based on median levels of free IxS, total HA, and free HA (Figure 1).

Moreover, no significant correlations were found between pCS and IAA levels with mortality in this study. A logistic regression model, describing the effect of each PBUT on death probability could explain 44% (Nagelkerke R^2^) of the variance in death and correctly classified 90.7% of cases. Patients with increased levels of free IxS and total and free HA were more likely to die within a 2-year follow-up period (*p* = 0.047, *p* = 0.013, *p* = 0.003, respectively). In patients who died during the follow-up period (n = 6, 11.11%, all of them in the second year), increased levels, compared to those remained alive, of total HA 6.42 (4.54–9.08) vs. 2.75 (1.46–4.49) mg/dL (*p* = 0.007); free HA 4.79 (2.47–6.4) vs. 1.36 (0.58–2.43) mg/dL (*p* = 0.004); free IxS 0.29 (0.14–0.42) vs. 0.13 (0.08–0.24) mg/dL (*p* = 0.045); total pCG 0.35 (0.25–0.74) vs. 0.19 (0.08–0.37) mg/dL (*p* = 0.045); free pCG 0.33 (0.23–0.65) vs. 0.18 (0.08–0.33) mg/dL (*p* = 0.042), were found, respectively (Figure 2). However, in these patients, increased levels of total pCS 1.34 (1.13–1.62) vs. 1.21 (0.8–1.68) mg/dL and free pCS 0.11 (0.08–0.25) vs. 0.08 (0.05–0.12) were also found, but were not statistically significant (*p* = 0.457, *p* = 0.16, respectively).

Regarding cardiovascular events, our study pointed out the potential importance of CMPF. Patients who experienced one or more cardiovascular events during follow-up (n = 11) had a higher concentration of CMPF [0.39 (0.18–0.85) vs. 0.15 (0.1–0.29) mg/dL (*p* = 0.015)] (Figure 3).

PBUT levels were not different among patients on HD with and without severe infections during the 2-year follow-up.

## 3. Discussion

In this study, we tried to investigate the possible association between immune alterations in HD, including increased serum PBUT levels and phenotypic changes in lymphocytes, and the morbidity and mortality of these patients. During a two-year follow-up, we found mortality in patients on HD was associated with increased levels of total and free HA, free IxS, and total and free pCG. Also, patients on HD with reduced naïve CD4 subpopulations were more likely to die within the next 2 years. However, multiple regression analysis showed that increased levels of free and total HA and free IxS were the only independent factors associated with increased risk of death, independently of age, dialysis vintage, and immune-phenotypic changes.

Results from previous studies have revealed rather controversial results. Some studies indicated an association between free serum levels of pCS and total IxS, with elevated all-cause mortality and also cardiovascular mortality in individuals undergoing HD [14,15]. Furthermore, elevated total pCS levels, correlated with cardiovascular events, including death from cardiac cause, myocardial ischemia, non-fatal myocardial infarction, ischemic stroke, peripheral vascular disease, and all-cause mortality in HD patients over 65 years of age [16]. Conversely, in the same population, the plasma concentrations of kynurenic acid, IxS, IAA, pCG, and HA did not demonstrate any association with either all-cause mortality or cardiovascular events [17].

HA is a metabolite that results from the hepatic glycine conjugation of benzoic acid. Benzoic acid can be produced by metabolic pathways of the intestinal microbiota, thence the ingestion of vegetal source foods (berries, milk, dairy products), abundant in polyphenols, specifically, chlorogenic acids or epicatechins. Moreover, it can be included in the foods as a preservative (beverages, industrial foods) [18,19]. Epicatechins can be found in tea, berries, broad beans, and barley, while chlorogenic acids in tea and coffee [18,20]. These two pathways (first related with epicatechins and second with chlorogenic acids), result in the synthesis of benzoic acid from the intestinal microbiome. This is absorbed in the circulation, and mainly in the liver (and less in the kidney) is converted to HA, after glycine conjugation [21,22]. Another pathway, is concerning the reduction in phenylalanine to phenylpropionic acid by intestinal microbiota, then is absorbed in the circulation and finally is converted in the liver to HA, through acyl-Coenzyme-A dehydrogenase β-oxidation [23]. So, HA is derived from the catabolism of dietary polyphenols of plant-based foods (including vegetables, fruits, coffee, tea) by the gut microbiota [24]. Excretion of HA is compromised in age-related conditions, such as cognitive impairments, sarcopenia, and hypomobility [24], and it has been proposed as a hallmark of aging and may have detrimental effects on both kidney and the brain [18]. Despite its low molecular weight of 179.2 Daltons, HA exhibits similar kinetics to those of larger molecules, due to its high protein binding, primarily to albumin, and demonstrates low clearance through dialysis [25,26]. Human serum albumin has two major regions of ligand binding. The HA binds with human serum albumin in both binding sites, with a greater affinity for site II, through hydrogen bonding, electrostatic, and hydrophobic interplay. As a result, there is low efficiency of removal of HA by dialysis [26].

IxS is originated from the breakdown of tryptophan by microbes within the colon. As a small molecule which is highly bound to plasma proteins (more than 90%), its clearance by HD is limited, leading to its peripheral blood accumulation [27]. Increased IxS levels are linked to cardiovascular damage in uremic patients, with underlying mechanisms being the contribution to endothelium dysfunction, vascular oxidative stress, inordinate inflammatory responses, calcification, and thrombosis [28]. PCS is a product of bacterial metabolism in the gut, and is associated with a plethora of biological adverse outcomes, including contribution to mechanisms that comprise renal (such as toxic effects on renal tubular cells) and cardiovascular damage [29]. Both IxS and pCS trigger arterial calcification in CKD and linked to endothelial dysfunction and oxidative stress [30]. CMPF, a furan fatty acid metabolite and a biomarker of fish intake, can interact with free oxygen radical, potentially provoking cell damage [31]. Kinetics of IxS, pCS, and CMPF in dialysis are similar of HA. HD with a high-flux membranes cannot effectively fend PBUTs, due to their increased serum albumin binding. IxS, pCS, and CMPF have protein binding proportion higher than 95% and removal ratio by HD lower than 35%. Clearance of IxS and pCS is meliorated by incrementing the diffusion of the free forms of PBUTs, with methods such as super-flux membranes, higher dialysate flow, hemodiafiltration, daily sessions of HD, and use of a sorbent to dialysate. Since CMPF has a binding ratio of 99–100% to albumin (higher than IxS and pCS), it cannot be cleared by conventional HD [32].

CMPF concentration was correlated with cardiovascular events in this study. Several PBUTs have been previously reported to demonstrate positive correlation with increased cardiovascular risk in patients on HD, although these findings are still controversial. Indoxyl sulfate was associated with symptomatic peripheral arterial disease, even after adjusting for established risk factors, but not with other major adverse cardiovascular events in HD [33]. HA is found to be correlated with left ventricular hypertrophy, a condition associated with elevated risk of death in this population [34]. In addition, HA contributes to endothelial dysfunction both in vitro and in vivo, resulting in progression of cardiovascular disease [25]. Serum total IxS levels were positively associated with all-cause mortality and severe infections [35], while plasma pCS may forecast the initial incidence of ischemic stroke in population on HD [36]. Furthermore, elevated PBUT levels are also found to correlate with other manifestations, including cognitive impairment, attributed to elevated IAA levels [37]. Conversely, results from other studies, indicated that pCS and IxS are not associated with cardiovascular outcomes, such as cardiac death and first cardiovascular event [38].

Regarding changes in lymphocytes and their subpopulations, the only subtype which showed close association with death, was that of naïve B cells (CD19+IgD+CD27-), found significantly reduced in HD patients who died during follow-up. Patients with kidney disease experience increased susceptibility to infection and have inadequate response to vaccination. A study showed that kidney dysfunction suspends germinal center response against T-dependent antigens, and germinal center B cells display greater apoptosis in kidney disease. HA causes loss of mitochondrial membrane potential, and results to higher apoptosis of germinal center B cells in a G-protein-coupled receptor 109A dependent manner. These B cells are reduced in mice with kidney disease, after infection with influenza virus, which is a mortality reason in patients with kidney disorders [39]. The absolute numbers of naïve B cells in HD patients are significantly reduced as compared with those in non-dialysis patients with stage V chronic kidney disease [40]. Memory B cells, switched memory B cells, naïve B cells, and IgG titers are independent risk factors for infection in HD patients, while infection is an ordinary issue and reason of mortality in this population [41]. Prior studies have demonstrated and highlighted the close association between lymphocytic alterations and death in patients on HD. Additionally, CD19+ B cell lymphocytopenia (<100 cells/μL) may serve as an independent predictor of all-cause and cardiovascular mortality in patients on HD. Some participants in the aforementioned study had received immunosuppression due to prior kidney transplantation, glomerulonephritis, autoimmune diseases, and other causes [42]. Reduced absolute number and percentage of T lymphocytes, low CD4+ naïve, and increased CD8+ central memory lymphocytes were linked to all-cause mortality, with a low count of naïve T lymphocytes being the sole independent factor [13]. Another study indicated that a decreased count of CD4+ naïve T lymphocyte was independently linked to cardiovascular events (coronary artery disease, congestive heart failure, stroke, peripheral arterial disease), while a low CD8+ naïve count was independently associated with severe infection episodes (infectious diseases necessitating intravenous antibiotics in hospital or emergency ward) in patients on HD [43]. The neutrophil to lymphocyte ratio may be predictive of mortality and cardiovascular events (myocardial infarction, ischemic heart disease, peripheral vascular disease, acute limb ischemia, mesenteric ischemia, stroke, deep venous thrombosis, pulmonary embolism), but not severe infections (infections with sepsis and/or requiring hospitalization and respiratory infection treated with antibiotics) in patients on HD, and demonstrated a positive association with the IxS [44]. Decreased memory B cells are irrespectively correlated with an increased risk of infections, particularly respiratory infections, in patients on HD [41].

A major limitation of this study is the small number of participants, both patients and control group. Secondly, due the low number of the clinical events we cannot surely reckon for other confounders in the analysis. The fact that the statistical analysis is based on a small sample and the number of deaths is limited restricts the ability to adjust for confounders and there is a potential lack of statistical power of the results.

Interestingly though, patients with increased morbidity were excluded, namely those with diabetes mellitus and/or active malignancy, as parameters that potentially affect lymphocytes’ phenotype and function. Patients with history of diabetes mellitus were excluded from this study, as this condition may have possible alterations in immunological status in HD patients. Specifically, patients with diabetes mellitus undergoing HD for half a year displayed an increase in CD3+CD8+, natural killer cells, CD4+CD28 null, and CD8+CD28 null [45]. Accordingly, our results are not representative of the whole cohort of the HD population, but rather of those non-diabetic patients being in a stable condition on HD. Furthermore, since it is an observational study, we could not prove causality of the relation between serum PBUT levels and clinical events but can simply describe that correlation and the interpretation of the results is appropriately cautious. It is essential to be executed other studies. Further investigation is required to reveal the mechanisms of the correlation between PBUTs and lymphocytes’ phenotype and to examine potential therapeutic suggestions.

## 4. Materials and Methods

### 4.1. Study Design

This is a prospective observational study, conducted in the First Department of Nephrology of Aristotle University of Thessaloniki, in collaboration with the Department of Immunology, General Hospital “Hippokratio”, Thessaloniki, Greece, from April 2022 to March 2024. Adult patients on thrice weekly chronic maintenance HD for at least 6 months were eligible for the study. Patients with history of diabetes mellitus, recent infection or vaccination (last 3 months), systematic autoimmune disease or recent medication with immunosuppressive drugs (last 12 months), and active malignancy or history of malignancy during the last 5 years were excluded from the study, as these conditions may have been associated with potential alterations in immunological status. All patients provided informed consent prior to enrollment and demographic and clinical data were recorded. For the analysis of immune phenotype, total blood samples were collected before the initiation of a mid-week dialysis session, and analyzed with flow cytometry as described below. At the same time point, plasma was collected, after centrifugation of whole blood in Ethylene diamine tetraacetic acid tubes for 10 min at 2095× *g*, and preserved in −60 °C for evaluation of PBUT levels with ultra-performance liquid chromatography. Subsequently, the patients were observed for a 24-month follow-up period and future clinical events were recorded.

Informed consent was obtained from all subjects involved in the study. The study was conducted in accordance with the Declaration of Helsinki and approved by the Institutional Review Board (or Ethics Committee) of the Medical School of the Aristotle University of Thessaloniki (protocol code 134/2023, 10 May 2023). All research activities were performed with coded, pseudonymized tissue samples and data.

### 4.2. Clinical Events

During the 24-month follow-up period clinical events were recorded, including cardiovascular ones, hospitalization due to infection, respiratory infection, and all-cause mortality. Cardiovascular events were defined as a composite of the first occurrence of congestive heart failure, acute myocardial infarction, coronary heart disease, atrial fibrillation, and peripheral arterial disease during the follow-up period. Hospitalization due to infection was defined as a composite of the occurrence of different kinds of infections (bacteremia, pneumonia and other pulmonary infection, urinary tract infection, soft tissue infection, vascular access infection, osteomyelitis, viral or fungal infection) that required hospitalization, during the follow-up period. Respiratory infection was defined as bacterial, viral or unknown origin or etiology infection of the respiratory tract that required treatment (such as antibiotics) during the follow-up period. All-cause mortality was defined as a composite of recorded causes of death (cardiac arrest, cardiovascular disease, infection, unknown, others) during the follow-up period.

### 4.3. Laboratory Methods

#### 4.3.1. Analysis of PBUTs

The blood preparation for total PBUT concentration was performed by ultra performance liquid chromatography and has been described in details previously [46]. The detailed explanation of the uremic toxins analysis process can be found in the Appendix A.

#### 4.3.2. Analysis of Lymphocytes

Whole blood samples were immediately transferred to the laboratory, where kept at room temperature until processing, for no longer than 12 h. Blood samples were stained with the following conjugated antibodies and analyzed with flow cytometry, according to manufacturer’s recommendations, as described previously [46]. Percentages and counts of T cell subpopulations were determined according to the expression of surface markers as following: early differentiated cells (recent thymic emigrants, naïve), memory cells (central memory, effector memory), advanced differentiated senescent cells [effector memory re-expressing CD45RA (EMRA)], and terminally differentiated senescent cells (EMRACD28-). B cells were classified as naïve, IgM memory, switched memory, and double negative. The detailed explanation of flow cytometry analysis process can be found in the Appendix A.

In the same of immune profile and PBUT evaluation, additional laboratory parameters: complete blood count, serum urea (enzymatic method, in mg/dL), serum creatinine (enzymatic kinetic method, in mg/dL), parathyroid hormone (chemiluminescent microparticle immunoassay method, in pg/mL), serum phosphorus (colorimetric method, in mg/dL,), serum calcium (colorimetric method, in mg/dL), serum lactate dehydrogenase (enzymatic kinetic method, in IU/L), serum albumin (colorimetric method, in g/dL), ferritin (chemiluminescent microparticle immunoassay method, in ng/mL), C-reactive protein (immunostaining method, in mg/L), and lipids [serum cholesterol (enzymatic method, in mg/dL), serum triglycerides (enzymatic method, in mg/dL), serum high-density lipoprotein cholesterol (enzymatic method, in mg/dL), and serum low-density lipoprotein cholesterol (enzymatic method, in mg/dL)].

### 4.4. Statistical Analysis

The statistical processing and analysis of the data was performed with the statistical package for social sciences (SPSS) version 27 IBM Corp, Armonk, NY, USA, for Windows. The level of statistical significance (*p*) was set below 0.05. The qualitative variables were described using absolute (n) and relative frequency (%). Kolmogorov–Smirnov and Shapiro tests were used to estimate the normality distribution of continuous variables. Normally distributed parameters were expressed as mean ± standard deviation (SD), while non-normally distributed parameters were expressed as the median and range. A Mann–Whitney U test was performed to estimate the differences in parameters between two groups (patients and control group, patient who experienced or not a clinical event), and Spearman’s correlation test was used to assess the correlations between non-parametric variables. Spearman’s coefficient and multiple regression analysis were performed to estimate the correlation between PBUT serum levels and clinical events. Kaplan–Meier and Cox regression model analysis were performed for future clinical events of patients on HD.

## 5. Conclusions

In patients on HD, increased levels of PBUTs, mainly free and total HA, were associated with increased risk of death within two years, independently of age, dialysis vintage, and immune-phenotypic changes. Patients on HD who died in the follow-up period had a significantly reduced count of naïve B cells compared to those remained alive. Further studies are needed to investigate methods of efficient removal of HA within HD, strategies for a lower production of HA by gut microbiota and examination of dietary changes for reduction in HA intake without losing the benefits of plant-based foods, such as dietary fibers. The causality of the relation between serum PBUT levels, lymphocytes’ phenotype, and clinical events cannot be inferred from this study, since it is an observational one.

## Figures and Tables

**Figure 1 ijms-26-10376-f001:**
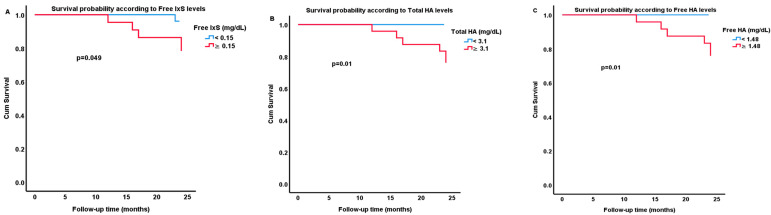
Kaplan–Meier curves of estimated survival probability for 2 different groups of patients on hemodialysis, according to protein bound uremic toxins serum levels [indoxyl sulfate (IxS) (**A**), total (**B**), and free (**C**) hippuric acid (HA)].

**Figure 2 ijms-26-10376-f002:**
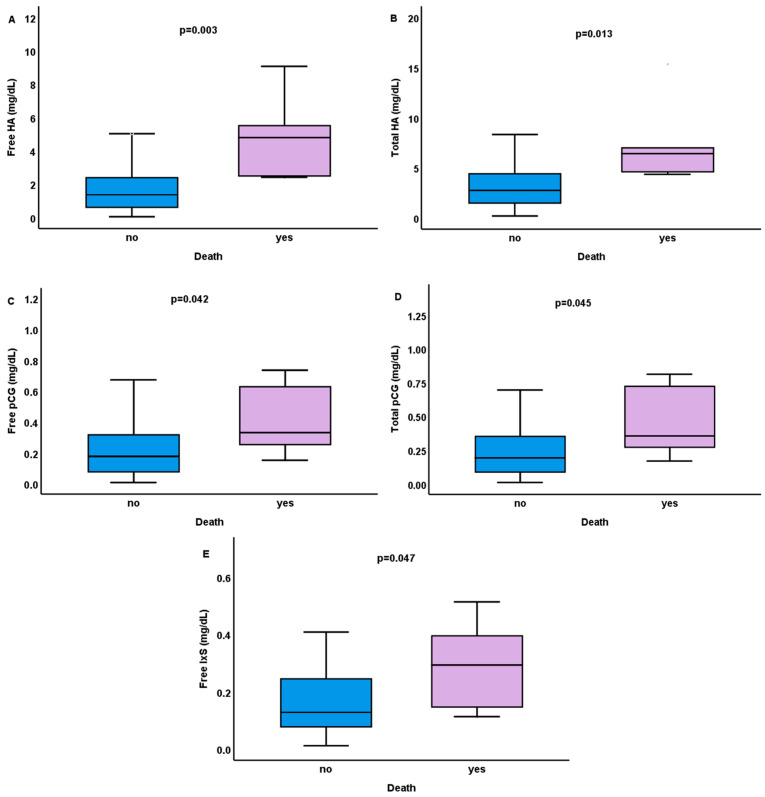
Differences in serum levels of free (**A**) and total (**B**) hippuric acid (HA), free (**C**) and total (**D**) p-cresyl glycuronide (pCG), and free indoxyl sulfate (IxS) (**E**) in patients on hemodialysis who died during 2-year follow-up period.

**Figure 3 ijms-26-10376-f003:**
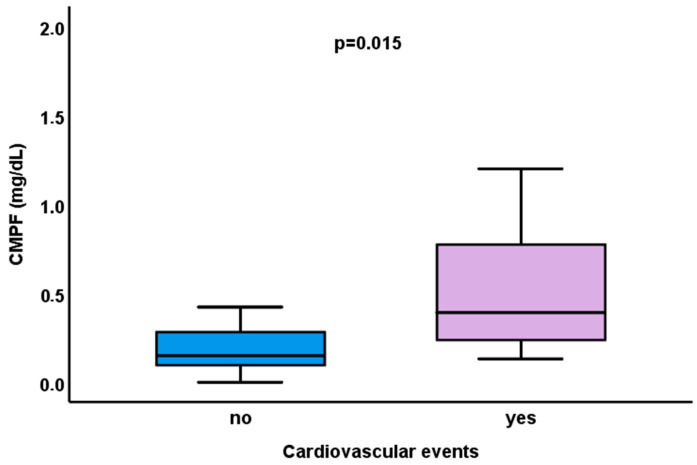
Differences in serum levels of 3-carboxy-4-methyl-5-propyl-2-furanpropionate (CMPF) in patients on hemodialysis who experienced a cardiovascular event during 2-year follow-up period.

**Table 1 ijms-26-10376-t001:** Dialysis related details, comorbidity, and causes of end-stage kidney disease (ESKD) in patients.

Parameters	Patients
Dialysis-related details	
Online hemodiafiltration/Hemodialysis	44.4%/55.6%
Duration of dialysis’ session (min)	240 (240–240)
Kt/V	1.45 (1.31–1.56)
Fistula/Central venous catheter	79.2%/20.8%
Comorbidity	
Arterial hypertension	66.7%
Dyslipidemia	31.5%
Hypothyroidism	11.1%
Peripheral arterial disease	9.3%
Coronary heart disease	7.4%
Atrial fibrillation	7.4%
Stroke	7.4%
Primary cause of ESKD	
Primary glomerulonephritis	29.6%
Not known	27.8%
Obstructive uropathy	16.7%
Autosomal dominant polycystic kidney disease	13%
Hypertensive kidney disease	5.5%
Alport syndrome	3.7%
Other	3.7%

**Table 2 ijms-26-10376-t002:** Laboratory parameters of patients and control group. *p* values are referred to differences between patients and control group.

Parameters	Patients	Control Group	*p*
Whole blood count			
Hemoglobin (g/dL)	11.8 ± 0.9	13.7 ± 1.1	<0.001
Hematocrit (%)	36.1 ± 3.1	40.7 ± 3.1	<0.001
White blood cells (cells/μL)	7100 (5500–8325)	6200 (5300–7100)	0.046
Neutrophils (cells/μL)	4550 (3475–5500)	3400 (2700–4200)	0.001
Lymphocytes (cells/μL)	1400 (1175–1800)	2100 (1600–2500)	<0.001
Platelets (10^3^/μL)	227 (184.2–265)	216 (199–245)	0.729
Urea (mg/dL)	127 (111.5–151.5)		
Creatinine (mg/dL)	9.4 (7.3–10.9)		
Parathyroid hormone (pg/mL)	206.5 (106.2–370.7)		
Phosphorus (mg/dL)	4.3 (3.7–5)		
Calcium (mg/dL)	9.1 (8.8–9.3)		
Albumin (g/dL)	4.1 (3.9–4.3)		
C-reactive protein (mg/L)	2.3 (1.4–4.2)		

**Table 3 ijms-26-10376-t003:** Patients’ and control group’s serum concentration levels of protein bound uremic toxins. *p* values are referred to differences between patients and control group.

Parameters	Patients	Control Group	*p*
Total IxS * (mg/dL)	2.21 (1.27–3.34)	0.06 (0.04–0.09)	<0.001
Free IxS * (mg/dL)	0.15 (0.09–0.27)	0.0004 (0.0004–0.0004)	<0.001
Total pCS * (mg/dL)	1.25 (0.84–1.66)	0.07 (0.04–0.13)	<0.001
Free pCS * (mg/dL)	0.09 (0.06–0.13)	0.004 (0.004–0.005)	<0.001
Total pCG * (mg/dL)	0.22 (0.09–0.39)	0.001 (0.0013–0.0017)	<0.001
Free pCG * (mg/dL)	0.2 (0.08–0.35)	0.002 (0.0011–0.0017)	<0.001
Total HA * (mg/dL)	3.1 (1.66–5.37)	0.1 (0.04–0.2)	<0.001
Free HA * (mg/dL)	1.48 (0.7–2.8)	0.03 (0.03–0.04)	<0.001
Total IAA * (mg/dL)	0.12 (0.09–0.167)	0.03 (0.02–0.03)	<0.001
Free IAA * (mg/dL)	0.04 (0.03–0.05)	0.006 (0.0056–0.0069)	<0.001
CMPF * (mg/dL)	0.17 (0.1–0.38)	0.07 (0.035–0.144)	<0.001

* indoxyl sulfate (IxS); p-cresyl sulfate (pCS); p-cresyl glucuronide (pCG); hippuric acid (HA); in-dole-3-acetic acid (IAA); 3-carboxy-4-methyl-propyl-2-furanpropanoic acid (CMPF).

**Table 4 ijms-26-10376-t004:** Differences in T and B lymphocyte subpopulations between patients and control group. *p* values are referred to differences between patients and control group.

Parameters	Patients	Control Group	*p*
White blood cells (cells/μL)	7100 (5500–8325)	6200 (5300–7100)	0.046
CD4+ (cells/μL)	679.5 (483–862.2)	999 (786–1237)	<0.001
CD8+ (cells/μL)	377 (261.7–531.5)	451 (296–746)	0.13
CD19+ (cells/μL)	91 (52.2–131.2)	248 (163–388)	<0.001
Naïve (IgD+CD27-) (cells/μL)	56.5 (27.7–96.2)	144 (91–258)	<0.001
IgM memory (IgD+CD27+) (cells/μL)	5 (3–10.2)	23 (11–32)	<0.001
Switched memory (IgD-CD27+) (cells/μL)	13 (7.7–18.5)	35 (22–61)	<0.001
Double negative (IgD-CD27-) (cells/μL)	7 (4.7–12)	26 (14–43)	<0.001

## Data Availability

The datasets used and analyzed during the current study are available from the corresponding author on reasonable request.

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
