# Peer review of "Lymphocyte Phenotypes and Protein-Bound Uremic Toxins as Determinants of Clinical Outcomes in Hemodialysis Patients"

_ijms, 2025, doi:10.3390/ijms262110376_

Round 1

Reviewer 1 Report

Comments and Suggestions for Authors

- At the beginning, it should be noted that the significantly high degree of text overlap confirmed via iThenticate of 39% is significantly high for original research and it is necessary to redesign the interpretation of references because this raises suspicions that it is either plagiarism or poor interpretation of references!

1. It is necessary to interpret the result in accordance with the parameters of dialyzability:
a) phenotype according to Kt/V?
b) phenotype according to residual renal function?
c) phenotype according to the type of dialysis treatment and dialysis membrane surface area?

2. The methodology lacks a way to determine standard analyses?

3. How is the determination of a standard cranial image, and how are biochemical analyses (devices) not just a technique?

4. Why is DM missing in primary kidney disease and comorbidities as the most common cause of ESKD?

5. How is the low % of cardiovascular comorbidity and 5% of patients with HTN explained?

6. Should the results be interpreted with total proteins and other fractions, not just albumin?

Now, because of all of the above, I would reject the paper and suggest that it be edited, changed, and then resubmitted to the IJMS!

Author Response

Response to Reviewer 1

Thank you very much for taking the time to review this manuscript. Please find the detailed responses below and the corresponding revisions/corrections highlighted changes in the resubmitted files.

3. Point-by-point response to Comments and Suggestions for Authors

Comment 1: - At the beginning, it should be noted that the significantly high degree of text overlap confirmed via iThenticate of 39% is significantly high for original research and it is necessary to redesign the interpretation of references because this raises suspicions that it is either plagiarism or poor interpretation of references!

Response 1: Thank you for pointing this out. We changed the text in the Introduction and in the Discussion, according to your suggestion, which improved the overall image of the article. We highlighted the changes in the resubmitted file.

Comment 2: 1. It is necessary to interpret the result in accordance with the parameters of dialyzability: a) phenotype according to Kt/V? b) phenotype according to residual renal function? c) phenotype according to the type of dialysis treatment and dialysis membrane surface area?

Response 2: We thank the reviewer for this valuable comment. Therefore, we added this in the last paragraph of the 3.2. Section (PBUT and peripheral lymphocytic phenotype of patients on HD) in Results (page number 6, second paragraph, lines 199-209), to emphasize this point, according to your recommendations. We highlighted the changes in the revised file.

“[Patients treated with online hemodiafiltration had slightly higher total white blood cells, neutrophils and lymphocytes compared to those on conventional HD [7150 (5775-8550) vs. 7000 (5275-7600) cells/μL (p=0.3), 4650 (3525-5875) vs. 4400 (3475-5400) cells/μL (p=0.47), 1500 (1325-1800) vs. 1400 (1050-1825) cells/μL (p=0.346), respectively], but these results were not statistically significant. Patients with and without residual renal function did not have statistically significant changes in the total numbers of white blood cells, neutrophils and lymphocytes [6900 (5450-7800) vs. 7100 (5600-8500) cells/μL (p=0.619), 4500 (3400-5200) vs. 4600 (3750-5850) cells/μL (p=0.267), 1600 (1300-1900) vs. 1400 (1000-1750) cells/μL (p=0.105), respectively] or in the lymphocyte phenotype. Additionally, lymphocytes did not correlated significantly with other parameters of dialyzability, such as Kt/V or dialysis membrane surface area.]”.

Comment 3: 2. The methodology lacks a way to determine standard analyses?

Response 3: Thank you for this comment. We discuss about the analysis of the protein-bound uremic toxins and the lymphocytes in the Supplementary Materials.

Comment 4: 3. How is the determination of a standard cranial image, and how are biochemical analyses (devices) not just a technique? 4. Why is DM missing in primary kidney disease and comorbidities as the most common cause of ESKD?

Response 4: Thank you very much for your comment, which actually was also our concern. We excluded patients with possible alterations in the immunological phenotype, such as those with diabetes mellitus. Diabetes mellitus itself, is proved to influence lymphocyte profile, especially immunosenescent profile, and that was the main reason we excluded these patients, in order to avoid the effect of diabetes, and look clearer to the association between uremia and immunological changes. Our study population cannot be representative the whole hemodialysis population, yet, we believe it covers a large percentage. Moreover, we added and modified these as limitations in the Discussion session (page number 11-12, lines 388-396). We highlighted the changes in the resubmitted file.

“[Interestingly though, patients with increased morbidity were excluded, namely those with diabetes mellitus and / or active malignancy, as parameters that potentially affect lymphocytes’ phenotype and function. Patients with history of diabetes mellitus were excluded from this study, as this condition may has a possible alterations in immunological status in HD patients. Especially, patients with diabetes mellitus undergoing HD for half a year, displayed an increase in CD3+CD8+, natural killer cells, CD4+CD28 null and CD8+CD28 null. Accordingly, our results are not representative of the whole cohort of HD population, but rather of those non-diabetic patients being in a stable condition on HD.]”.

Comment 5: 5. How is the low % of cardiovascular comorbidity and 5% of patients with HTN explained?

Response 5: Thank you for this comment. “[The low percentage of cardiovascular comorbidity may be explained due to the fact that patients with high comorbidity were excluded, such as those with diabetes mellitus, and also, our patients were relatively young.]”. We accordingly modified the text in the first paragraph of the 3.1. Section (Patients’ characteristics) in Results (page number 4, third paragraph, lines 170-172). 66.7% of the patients had arterial hypertension and 5.5% of the patients had hypertensive kidney disease as primary cause of end stage kidney disease. We changed the term “hypertension” with the term “hypertensive kidney disease” at Table 1. The majority of patients had primary glomerulonephritis as primary cause of end stage kidney disease, since our center has a plethora of patients with glomerulonephritis from Northen Greece. We highlighted the changes in the revised file.

Comment 6: 6. Should the results be interpreted with total proteins and other fractions, not just albumin?

Response 6: Thank you very much for this suggestion. The protein bound uremic toxins that were examined (indoxyl sulfate, p-cresyl sulfate, p-cresyl glycuronide, hippuric acid, indole-3-acetic acid, 3-carboxy-4-methyl-5-propyl-2-furanpropionate) are bound with human serum albumin. We did not correlate total proteins and other fractions with other parameters in the Results. We correlated the total and free serum levels of these uremic toxins with peripheral lymphocytic phenotype of patients on hemodialysis and with clinical events in this population.

We have revised the manuscript according to your suggestions.

Thank you very much for all your help.

Kind regards,

Maria Stangou

Professor in Nephrology AUTH

Hippokration Hospital

Thessaloniki, Greece

Reviewer 2 Report

Comments and Suggestions for Authors

Dear Authors,

The authors’ manuscript examines how protein-bound uremic toxins (PBUTs) and changes in lymphocyte subtypes in hemodialysis (HD) patients are associated with 2-year mortality and clinical events, which is of high clinical relevance. In particular, the clear demonstration of the association between hippuric acid (HA) and naïve B cells (CD19+IgD+CD27-) represents a novel finding. On the other hand, the study has a small sample size and is observational in nature, so causality cannot be inferred. The following points could be improved:

Statistical analysis and limitations should be explicitly stated. Specifically, the manuscript should clarify that the analysis is based on a small sample, that the number of deaths is limited, restricting the ability to adjust for confounders, and indicate the potential lack of statistical power to the readers.

Figures and tables need refinement. In particular, the Kaplan-Meier curves and PBUT concentration figures should be reorganized, and the presentation of p-values and χ² values should be standardized. Redundant p-value reporting in the text should also be simplified.

Discussion of the clinical relevance of PBUTs should be strengthened. The manuscript should provide a more detailed discussion of HA, including its intestinal metabolism, dietary influences, and the low efficiency of removal by dialysis, and clarify comparisons with other PBUTs such as IxS and pCG.

Interpretation of immune phenotypes should be expanded. The association between decreased naïve B cells and mortality risk should be discussed in the context of T cell subsets, existing literature, and potential mechanisms (e.g., impaired immune function and increased susceptibility to infection).

Clarity of writing should be improved. The manuscript should repeatedly emphasize that causality cannot be inferred from an observational study and ensure that the interpretation of results is appropriately cautious.

Author Response

Response to Reviewer 2 Comments

1. Summary

Thank you very much for taking the time to review this manuscript. Please find the detailed responses below and the corresponding revisions/corrections highlighted changes in the resubmitted files.

3. Point-by-point response to Comments and Suggestions for Authors

Comment 1: Dear Authors, The authors’ manuscript examines how protein-bound uremic toxins (PBUTs) and changes in lymphocyte subtypes in hemodialysis (HD) patients are associated with 2-year mortality and clinical events, which is of high clinical relevance. In particular, the clear demonstration of the association between hippuric acid (HA) and naïve B cells (CD19+IgD+CD27-) represents a novel finding. On the other hand, the study has a small sample size and is observational in nature, so causality cannot be inferred. The following points could be improved: Statistical analysis and limitations should be explicitly stated. Specifically, the manuscript should clarify that the analysis is based on a small sample, that the number of deaths is limited, restricting the ability to adjust for confounders, and indicate the potential lack of statistical power to the readers.

Response 1: Thank you for pointing this out. Therefore, we modified the text in the revised manuscript. This change can be found in the last paragraph of the Discussion (page number 11, third paragraph, lines 385-387, page number 12, first paragraph, lines 396-401). Moreover, we added this in the Conclusions (page number 12, second paragraph, lines 410-411), to emphasize this point. We highlighted the changes in the resubmitted file.

“[The fact that the statistical analysis is based on a small sample and the number of deaths is limited, restricts the ability to adjust for confounders and there is a potential lack of statistical power of the results.]”, “[Furthermore, since it is an observational study, we could not prove causality of the relation between serum PBUT levels and clinical events, but can simply describe correlation and the interpretation of the results is appropriately cautious. It is essential to be executed other studies. Further investigation is required to reveal the mechanisms of the correlation between PBUT and lymphocytes’ phenotype and to examine potential therapeutic suggestions.]” and “[The causality of the relation between serum PBUT levels, lymphocytes’ phenotype and clinical events cannot be inferred from this study, since it is an observational one.]”.

Comment 2: Figures and tables need refinement. In particular, the Kaplan-Meier curves and PBUT concentration figures should be reorganized, and the presentation of p-values and χ² values should be standardized. Redundant p-value reporting in the text should also be simplified.

Response 2: Thank you very much for your comment. We refined figured and tables, as you suggested. Additionally, we removed some p-values in the text which were reported at the Figures or Tables.

Comment 3: Discussion of the clinical relevance of PBUTs should be strengthened. The manuscript should provide a more detailed discussion of HA, including its intestinal metabolism, dietary influences, and the low efficiency of removal by dialysis, and clarify comparisons with other PBUTs such as IxS and pCG.

Response 3: Thank you for this comment. We provided a more detailed discussion of HA, including its intestinal metabolism, dietary influences, and the low efficiency of removal by dialysis in the Discussion (page number 9-10, lines 290-302, 309-312). We added a comparison of the kinetics in dialysis of HA and IxS, pCG, CMPF in the Discussion (page number 10, lines 324-332). We highlighted the changes in the revised file.

“[HA is a metabolite which results from the hepatic glycine conjugation of benzoic acid. Benzoic acid can be produced by metabolic pathways of the intestinal microbiota, thence the ingestion of vegetal source foods (berries, milk, dairy products), abundant in polyphenols, specifically, chlorogenic acids or epicatechins. Moreover, it can be included in the foods as a preservative (beverages, industrial foods). Epicatechins can be found in tea, berries, broad beans and barley, while chlorogenic acids in tea and coffee. These two pathways (first related with epicatechins and second with chlorogenic acids), result in the synthesis of benzoic acid from the intestinal microbiome. This is absorbed in the circulation, and mainly in the liver (and less in the kidney) is converted to HA, after glycine conjugation. Another pathway, is concerning the reduction of phenylalanine to phenylpropionic acid by intestinal microbiota, then is absorbed in the circulation and finally is converted in the liver to HA, through acyl-Coenzyme-A dehydrogenase β-oxidation.]” and “[Human serum albumin has two major regions of ligand binding. The HA binds with human serum albumin in both binding sites, comparatively more at site II, through hydrogen bonding, electrostatic and hydrophobic interplay. As a result, there is low efficiency of removal of HA by dialysis.]” and “[Kinetics of IxS, pCS and CMPF in dialysis are similar of HA. HD with a high-flux membranes cannot effectively fend PBUTs, due to their increased serum albumin binding. IxS, pCS and CMPF have protein binding proportion higher than 95% and removal ratio by HD lower than 35%. Clearance of IxS and pCS is meliorated by incrementing the diffusion of the free forms of PBUTs, with methods such as super-flux membranes, higher dialysate flow, hemodiafiltration, daily sessions of HD and use of a sorbent to dialysate. Since CMPF has a binding ratio of 99-100% to albumin (higher than IxS and pCS), it cannot be cleared by conventional HD.]”.

Comment 4: Interpretation of immune phenotypes should be expanded. The association between decreased naïve B cells and mortality risk should be discussed in the context of T cell subsets, existing literature, and potential mechanisms (e.g., impaired immune function and increased susceptibility to infection).

Response 4: Thank you very much for this comment. We added a more detailed discussion of these lymphocytes and the mortality in the Discussion (page number 11, second paragraph, lines 350-362), according to your suggestion. We highlighted the changes in the resubmitted file. As a result, from your comments 3 and 4, we added date from 9 more studies, compared to the original manuscript, so that the concept of the article is more understandable.

“[Patients with kidney disease experience increased susceptibility to infection and have inadequate response to vaccination. A study showed that kidney dysfunction suspends germinal center response against T-dependent antigens, and germinal center B cells display greater apoptosis in kidney disease. HA causes loss of mitochondrial membrane potential, and results to higher apoptosis of germinal center B cells in a G-protein–coupled receptor 109A dependent manner. These B cells are reduced in mice with kidney disease, after infection with influenza virus, which is a mortality reason in patients with kidney disorders. The absolute numbers of naïve B cells in HD patients are significantly reduced as compared with those in non-dialysis patients with stage V chronic kidney disease. Memory B cells, switched memory B cells, naïve B cells and IgG titers are independent risk factors for infection in HD patients, while infection is an ordinary issue and reason of mortality in this population.]”.

Comment 5: Clarity of writing should be improved. The manuscript should repeatedly emphasize that causality cannot be inferred from an observational study and ensure that the interpretation of results is appropriately cautious.

Response 5: Thank you for mentioning this specific limitation. We emphasize this point in the last paragraph of the Discussion (page number 12, first paragraph, 398-401) and in the Conclusions (page number 12, second paragraph, 410-411). We highlighted the changes in the revised file.

“[the interpretation of the results is appropriately cautious. It is essential to be executed other studies. Further investigation is required to reveal the mechanisms of the correlation between PBUT and lymphocytes’ phenotype and to examine potential therapeutic suggestions.]” and “[The causality of the relation between serum PBUT levels, lymphocytes’ phenotype and clinical events cannot be inferred from this study, since it is an observational one.]”.

We have revised the manuscript according to your suggestions.

Thank you very much for all your help.

Kind regards,

Maria Stangou

Professor in Nephrology AUTH

Hippokration Hospital

Thessaloniki, Greece

Round 2

Reviewer 1 Report

Comments and Suggestions for Authors

No